# The influence of UV-visible light, microwave radiation, argon laser, and heating and aging processes on silicone oil utilized as intravitreal implants: Experimental exposure with clinical correlation

**Rami Al-Dwairi**[1]*, **Ahmad. A. Ahmad**[2], **Abdelwahab Aleshawi**[1], **Qais M. Al-Bataineh**[2], **Areen Bani-Salameh**[2], **Ihsan A. Aljarrah**[2], **Seren Al Beiruti**[1], **Abdulrawof Alhabachi**[1]

1 Division of Ophthalmology, Department of Special Surgery, Faculty of Medicine, Jordan University of Science & Technology, Irbid, Jordan, 2 Thin Films and Nanotechnology Lab, Department of Physics, Jordan University of Science & Technology, Irbid, Jordan

* ramialdwairi@yahoo.com

## Abstract

### Purpose

The emulsification of silicone oil (SO) remains poorly understood. In the present study, we investigated the physical properties of unused pharmaceutical SO samples under various conditions. Moreover, clinical correlations with the patients' SO samples were assessed.

### Methods

Unused pharmaceutical ophthalmic SO samples and four explanted SO samples from previously vitrectomized patients with rhegmatogenous retinal detachment were analyzed. To assess the stability of SO, the samples were exposed to UV light, visible light, a green argon laser, microwave radiation, heat, and were aged. Following exposure, the samples were investigated using Fourier-transform infrared spectroscopy, optical transmittance and absorbance, and micro-viscometry measurements. Two patients underwent argon laser retinopexy while SO tamponaded the vitreous cavity postoperatively.

### Results

The physical properties of SO exposed to heat, UV light, microwave radiation, and aging did not change. However, SO irradiated by the green argon laser demonstrated a significant breakdown of atomic bonding and a transmittance loss. These results are consistent with the analyses of SO samples provided by patients. In those who underwent laser retinopexy, the SO exhibited clinical emulsification necessitating earlier removal, which was confirmed by physical tests.

**Data Availability Statement:** All relevant data are within the manuscript and its Supporting Information files.

**Funding:** The authors would like to acknowledge Jordan University of Science and Technology for supporting the research projects validated by the Deanship of Scientific Research (Funding ID: 20230529). The funders had no role in study design, data collection and analysis, decision to publish, or preparation of the manuscript.

**Competing interests:** The authors have declared that no competing interests exist.

## Conclusions

It may be postulated that patients exposed to the argon laser experienced more emulsification than the other two groups, leading to the earlier removal of the SO implants from the eyes. This investigation did not consider the biological effects of inflammatory biomarkers; however, it may provide valuable insights for optimizing the use of SO in ophthalmic surgery and other potential applications.

## 1. Introduction

Cibis et al. were the first to introduce silicone oil (SO) in ophthalmic surgery to replace vitreous tissue in treating retinal detachment (RD) [1]. SO was first used by Scott, Heimann, and many others across Europe as early as 1962 as an intraocular tamponade following vitreoretinal surgery [2, 3]. SO is utilized due to its natural properties, including its transparency, high surface tension, low density, wide viscosity range, chemical inertness, and high interfacial tension with water. It comprises a siloxane polymer (−R2Si−O−SiR2−, where R denotes the organic group) known as silicone. The silicones comprise an inorganic silicon-oxygen backbone chain (···−Si−O−Si−O−Si−O−···) with two organic groups linked to each silicon core [4]. In a subsequent assessment in 1994, SO was approved by the Food and Drug Administration as a retinal tamponade in the United States [5]. Subsequently, it became irreplaceable in treating complicated vitreoretinal cases despite other advancements in vitreoretinal surgery. Despite the different roles and indications of SO, particularly in the treatment of complex RD, its removal is generally recommended because long-term tamponade by SO can lead to a wide range of complications [6].

The relative indications for SO vary significantly depending on the surgeon's experience and the patient's choice. The most common indications are complicated RD due to proliferative vitreoretinopathy (PVR), giant retinal tears, trauma, or viral retinitis. Another indication included RD resulting from a high myopia-related macular hole, colobomatous RD, chronic and persistent macular hole, and chronic uveitis with hypotony [5, 7–11]. SO is also used to treat rhegmatogenous RDs (RRDs), such as macular hole-related RRDs and RRDs with extensive posterior breaks, where it is difficult to perform effective laser retinopexy [5, 12]. Other common indications include advanced diabetic retinopathy (tractional RD) and exudative RD caused by an intraocular tumor [5, 13–16]. Moreover, SO is considered an optimal choice for short-term tamponade in patients who experience difficulty maintaining the required head position for gas tamponade or in those who travel at high altitudes [5].

A longer duration of SO tamponade has been associated with the development of various complications, with variable incidences. These include uveitis, retinal reattachment failure, cataract formation, glaucoma, SO emulsification, epiretinal membrane formation, and keratopathy [6].

This study investigated the optical, physical, and chemical properties of unused pharmaceutical SO under different conditions. By exploring these properties, this study aimed to provide valuable insights for optimizing the use of SO in ophthalmic surgery and other potential applications. Untreated pharmaceutical SO samples were exposed to ultraviolet (UV) light (low and high wavelengths), visible light, thermal heat, microwave radiation, and an argon laser; additionally, they were aged. Moreover, similar physical and chemical characteristics were investigated for the four SO samples extracted from previously vitrectomized patients, and these findings were clinically correlated with the experimental results.

## 2. Methods

### 2.1 Patients and study design

After obtaining approval from the Institutional Review Board (IRB) at Jordan University of Science and Technology (number 11/136/2023), we prospectively exposed unused pharmaceutical ophthalmic SO samples to different conditions from 05/09/2023 to 28/12/2023 to investigate the stability of this ophthalmic form of SO under variable conditions within the eye. Exposure factors included ultraviolet (UV) light, visible light, argon laser, microwave radiation, heating, and aging. Following exposure, the physical and chemical properties of the samples were investigated.

To correlate these findings clinically, we analyzed the chemical and physical properties of four SO samples extracted from the vitrectomized eyes of four patients who underwent primary pars plana vitrectomy (PPV) for RRD. The patients were divided into two groups; each group consisted of two patients matched for age and sex, with different exposures to argon retinal laser retinopexy. Argon retinal laser retinopexy was applied while the retina was kept flat under the tamponade effect of SO, either due to insufficient endolaser administration during the primary PPV or due to the development of new tears during follow-up visits. Demographic and medical data of the patients were recorded, along with operative details. Furthermore, data on the postoperative course, the need for argon laser retinopexy, indications for the laser, visual outcomes, and retinal status after SO removal were collected.

Written informed consent was obtained from all the patients. This study was conducted in accordance with the ethical standards of the 1964 Declaration of Helsinki and its subsequent amendments. We confirmed that the privacy of the participants was preserved and that the data was anonymized and kept confidential. Ethical approval was obtained from the Institutional Board of the Jordan University of Science and Technology (fund ID: 11/163/2023). This research was performed at the Ophthalmology Division of King Abdullah University Hospital and the Department of Physics (Thin Films and Nanotechnology Laboratory) at the Jordan University of Science and Technology.

### 2.2 Perioperative setting and samples collection

A single consultant vitreoretinal surgeon performed primary PPV and SO removal using similar surgical guidelines, as described in our earlier experiments [7, 17]. Three 23-gauge sclerotomies were created, and trocars were inserted using a vitrectomy system (Combined Wide-Field Elite Pack, Bausch, and Lomb) with a biplanar single-step technique. The infusion pressure was set at 30 mmHg by connecting the inferior temporal trocar to the infusion system. Using a machine-driven vacuum, up to 2 mL of a pure, undiluted, unmixed SO sample was explanted while the vacuum pressure was set at 60 mmHg. At this point, the retina was carefully checked for redetachment. After the extraction of SO from the vitreous cavity, the SO sample was preserved in a dark, sterile plastic syringe. It was then transferred to a physics laboratory, preserved in a temperature-controlled room with low humidity.

Anterior chamber paracentesis was performed to obtain an aqueous humor sample, which was added to the unused SO samples to investigate the effect of the aging process. A microvitreoretinal blade was used to create the side port for the anterior chamber. Subsequently, aqueous fluid (0.5 mL) was aspirated using a small-gauge cannula.

### 2.3 Setting of exposure parameters and conditions

New "unused" pharmaceutical SO samples (Huile Silicone Purifie Csi, FCI, France) were procured to study the stability of SO under different parameters.

The first sample was exposed to heat at 40˚C for 1 h, using an oven at ambient conditions (under air, normal pressure) (TLI, Jordan).

The second and third samples were exposed to long- and short-wavelength UV light (260 and 360 nm, respectively) at a power of 16 watts and an adjustable distance of approximately 10 cm from the sample for 1 h at room temperature—software (Lagaay Worldwide Medical Supplier, Netherlands).

The fourth sample was exposed to a microwave, with a power of 16 watts, at an adjustable distance of approximately 10 cm for 1 h at room temperature (Fisher Scientific, USA).

The fifth sample was exposed to visible light (wavelength 550 nm) at 16 watts and an adjustable distance of approximately 10 cm for 1 h at room temperature (AZDENT, China).

Furthermore, the sixth sample was exposed to and treated with a green argon laser using an ophthalmic pan-retinal photocoagulation device in the ophthalmic clinic (Valon Lasers Oy, Merimiehenkuja 5, Finland). The wavelength of the laser was set to 532 nm, and the frequency of the mode laser was double that of Nd-YVO. The laser power used was 500 μW with a pulse duration of 0.1 s and a spot size of 400 μm.

Finally, another factor that could affect SO is aging or the duration of tamponade within the eye. To simulate the effect of aging on SO inside the eye, we mixed it with a balanced salt solution (BSS) and eye aqueous humor. The properties of these mixtures were studied after aging them for 6 and 12 months.

## 2.4 Physics and optical characterization

The chemical properties of the unused pharmaceutical SO samples, exposed to the above mentioned parameters, and the SO samples extracted from the vitrectomized eyes were investigated using Fourier-transform infrared spectroscopy (FTIR) (ALPHA, FTIR spectrometer, Bruker, USA) in the wavenumber range of 400–4,000 $cm^{-1}$. FTIR spectroscopy was used to identify the functional groups of SO and their vibrational bonding modes within the intermolecular atomic bands. FTIR spectra were recorded in transmittance mode from 400 to 4,000 $cm^{-1}$ at a resolution of 2 $cm^{-1}$. FTIR is used to obtain the infrared spectrum of the absorption or emission of a solid, liquid, or gas. It is an analytical technique used to identify organic, polymeric, and, in some cases, inorganic materials [18].

Furthermore, the optical transmittance and absorbance spectra were measured using ultraviolet-visible (UV-Vis) spectral scanning at wavelengths ranging from 200 to 800 nm at a resolution of 1 nm at room temperature using an "EMC-61PC-UV" spectrophotometer (Oxford Instrument, UK). The kinematic viscosity was measured using a micro-viscometer (HVROC-S, RheoSense).

## 3. Results

Polydimethylsiloxane (PDMS) with the chemical formula $[Si(CH_3)_2O]_n$, also known as SO, is the subject of our project. It contains a long chain of linked monomers, each containing a silicon atom sharing two oxygen atoms at the main wings and $CH_3$ molecules at the other two accessible peripherals. The long linear chain consists of repeated O = Si = O segments that must be strong enough to maintain a robust polymeric chain with a high molecular weight, as observed in SO used in ophthalmic practice.

### 3.1 Stability of SO under diverse conditions of UV light, visible light, microwave radiation, and heat (temperature)

Different parameters, including heat (temperature), microwave radiation, UV radiation, and visible light radiation, have been hypothesized to affect the human eye. First, the FTIR spectra

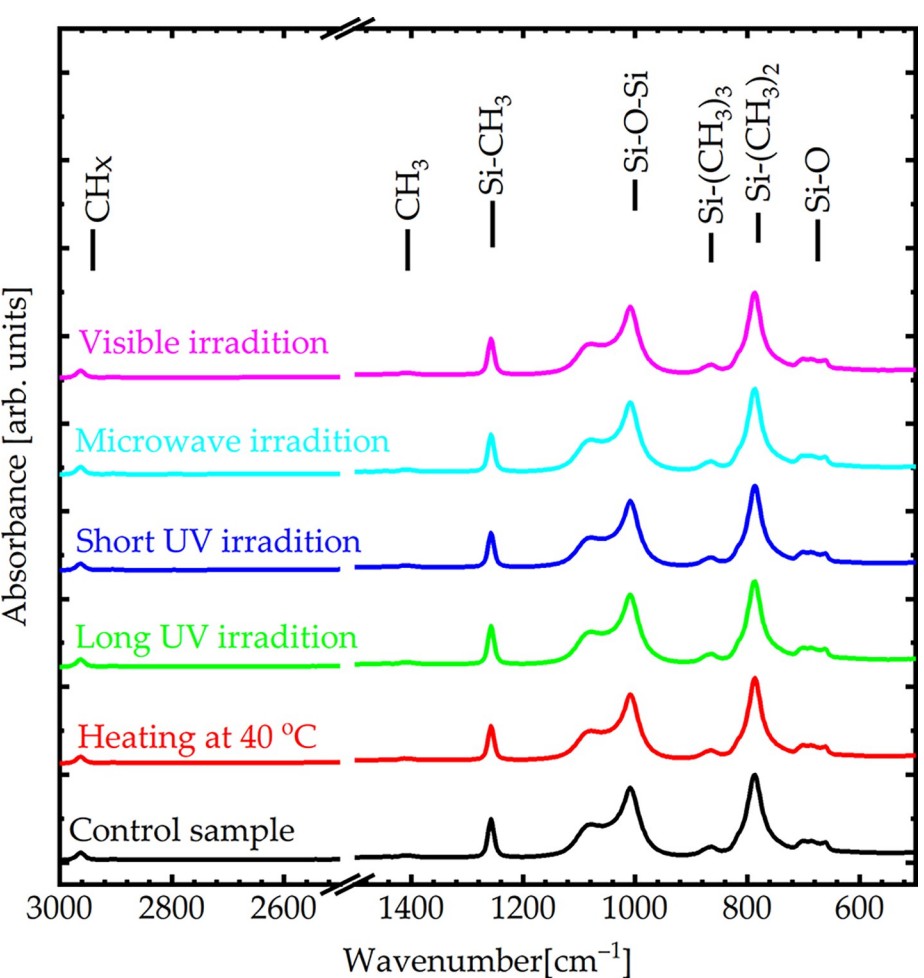

**Fig 1. FTIR spectra of SO under different parameters, including heat, UV radiation, visible light radiation, and microwave radiation.**

were used to investigate the effects of SO on the vibrational bands (Fig 1). SO exhibited different vibrational bands based on its chemical structure: Si–O (680 cm⁻¹), Si–(CH₃)₂ (790 cm⁻¹), Si–(CH₃)₃ (870 cm⁻¹), Si–O–Si (1,010 cm⁻¹), Si–CH₃ (1,260 cm⁻¹), CH₃ (1,405 cm⁻¹), and CHₓ (2,960 cm⁻¹). Si–O and Si–O–Si bonds are the main components of the skeleton. If they are broken, oxygen will have a chance to connect with other elements, such as C or H, and Si will also have an opportunity to bond with H. A closer examination of the Si–O and Si–O–Si bonds reveals that the intensity remained consistent across all samples, indicating that no bonds were broken and, therefore, no emulsification occurred. The FTIR spectra showed that the polymeric SO did not undergo degradation, intermolecular bond cleavage, or the formation of new bonds. The vibrational bands were unaffected by exposure to heat, UV radiation, visible light, and microwave radiation (Fig 1). Therefore, we conclude that the chemical structure of SO was stable under these conditions.

Moreover, the optical absorbance and transmittance of SO in the UV and visible regions, under different conditions (heat, UV radiation, visible light radiation, and microwave radiation), were measured using UV-Vis spectroscopy (Fig 2). In the visible and UV regions, the absorbance and transmittance values of SO under different parameters (heat, UV radiation,

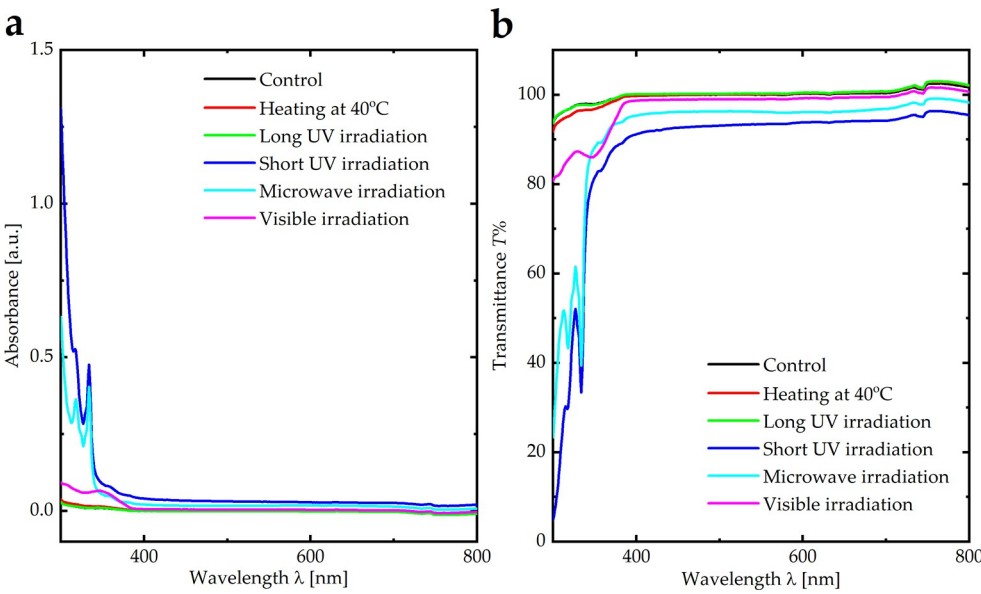

**Fig 2. (a)** Absorbance and **(b)** transmittance spectra of SO under different parameters, including heat, UV radiation, visible light radiation, and microwave radiation.

visible light radiation, and microwave radiation) were similar to those of the control sample of SO (Table 1), indicating that its optical properties were stable under these parameters.

The viscosity of the control SO was 5,509 mPa.s. The viscosity values of SO under different conditions (heat, UV, visible light, and microwave radiation) varied slightly, indicating that the viscosity readings were stable under these conditions (Table 1).

### 3.2 Stability of SO after aging (timing)

Another factor that may affect SO's physical and chemical characteristics is aging (timing). We studied the properties of SO after aging it for 6 and 12 months at a fixed temperature of approximately 34–35˚C, resembling the temperature of the mid-vitreous, which is typically 33.8 ± 1.1˚C (Fig 3) [19]. A closer examination of the vibrational bands of SO after aging for 6 and 12 months revealed a similar intensity, peak position, and linewidth, indicating no broken bonds and, therefore, no emulsification (Fig 3A). Additionally, the optical absorbance of SO in the UV and visible regions after aging for 6 and 12 months revealed that the properties of aged SO remained almost identical to those of the control sample, indicating that the optical properties were stable with aging (Fig 3B). The viscosity of the control SO/BSS/aqueous humor

**Table 1. Viscosity and transmittance values of SO under different parameters, including heat, UV radiation, visible light radiation, and microwave radiation.**

| Parameter | Transmittance T% at 550 nm | Viscosity [mPa.s] |
|---|---|---|
| Control | 99.98 | 5,509 |
| Heating at 40˚C | 99.97 | 5,767 |
| Long UV irradiation | 99.95 | 5,618 |
| Short UV irradiation | 93.37 | 5,313 |
| Microwave irradiation | 96.14 | 5,668 |
| Visible irradiation | 98.95 | 5,635 |

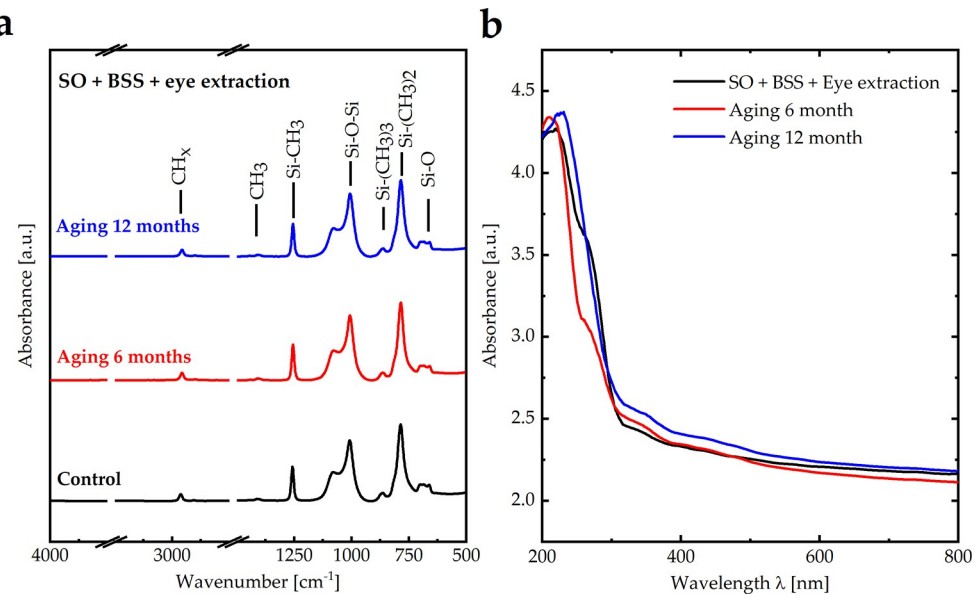

**Fig 3.** **(a)** FTIR and **(b)** UV-vis absorbance spectra of SO after aging for 6 and 12 months.

mixture was 7,175 mPa.s. The viscosity values of this mixture after aging for 6 and 12 months were 7,102 and 7,131 mPa.s, indicating that the viscosity was stable with the aging (timing) process.

### 3.3 Stability of SO under green argon laser exposure

One factor that may induce emulsification is the effect of argon laser exposure in vitrectomized patients with an SO tamponade. Fig 4 shows the FTIR spectra of SO before and after argon laser irradiation. Radiation of SO by a green argon laser demonstrated an apparent effect on the Si–O and Si–O–Si bonds, indicating broken bonds and emulsification in the polymer. In addition, the $Si(CH_3)_2$ band was principally bonded to the control sample. After argon laser radiation, the intensity and the linewidth of this band were affected by argon laser radiation, indicating a breakdown in the bond. All bands shifted to higher wavenumbers, indicating energy loss during bonding. This suggests that the argon laser radiation may significantly negatively affect the building structure of the SO tamponade.

The broken bonds, especially the Si—O, and Si—O—Si bonds, were definite evidence of a reduction in the polymer's molecular weight. This indicates that the laser-exposed samples had reduced viscosity, density, surface tension, and specific gravity.

Furthermore, the optical absorbance and transmittance of SO in the UV and visible regions under argon laser radiation were measured using UV-Vis spectroscopy (Fig 5). In the visible region, the transmittance of SO at 550 nm decreases from approximately 95% to 65% under argon laser radiation, indicating that the optical properties are unstable under argon laser radiation. In addition, the transmittance of SO over the UV and visible ranges decreased dramatically, and the absorbance increased, indicating the production of free radicals in the polymer chain, contributing to the increase in absorbance and decrease in transmittance.

The viscosity of the control SO sample was 5,394 mPa.s, which decreased to 3,309 mPa.s after argon laser radiation, indicating a decomposition in the sample's molecular weight.

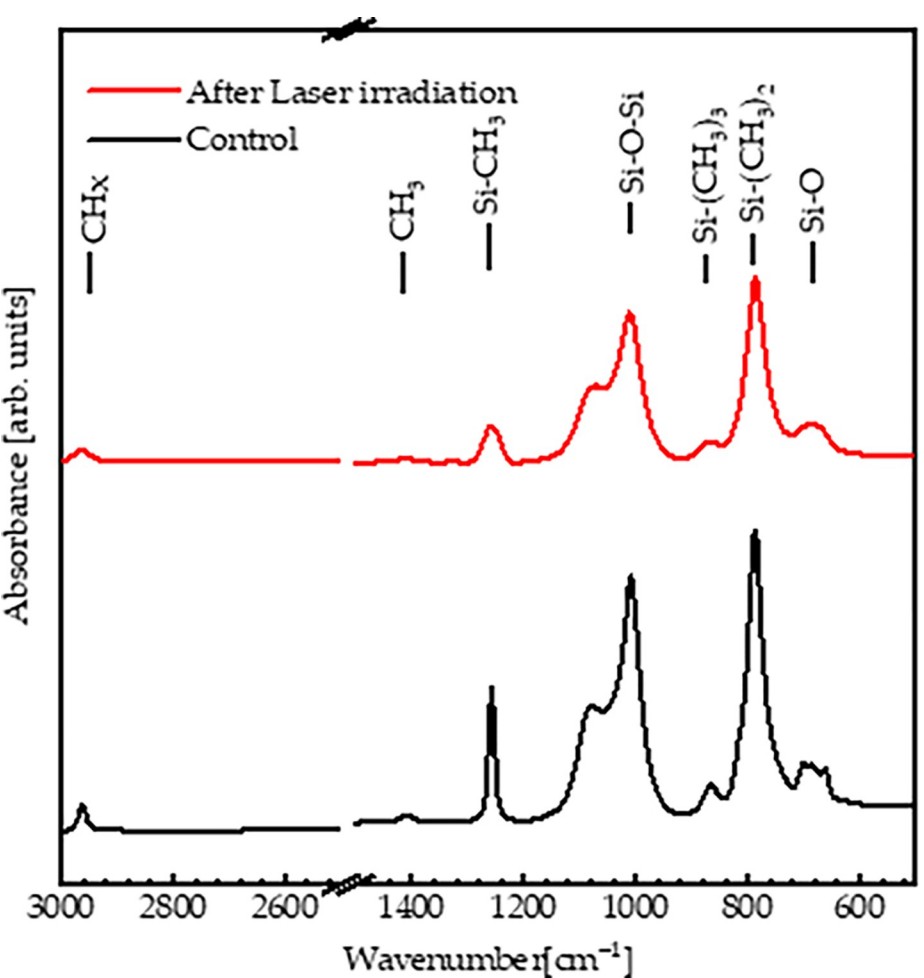

**Fig 4. FTIR spectra of pure SO and SO under argon laser radiation.**

### 3.4 Clinical cases involving exposure to argon laser

**3.4.1 Clinical, demographic, and operative characteristics.** All patients underwent primary PPV for RRD. Table 2 summarizes their demographics, and Table 3 summarizes their clinical outcomes.

*Patient 1 versus patient 2.* Patient 1 was a 47-year-old female; she presented with symptoms suggestive of RRD in the left eye. On examination, superior temporal RRD was confirmed. Urgent PPV was arranged, and the operative findings included superior-temporal RRD from 2–7 o'clock, with a tear located at the 2 o'clock position and peripheral lattice and cystic changes. The macula was removed. Intraoperatively, an endolaser was applied around the tear and extended 360˚. SO was injected, achieving a good fill. After 19 months, SO removal with cataract extraction was performed in the left eye without complications. One month later, the patient presented with a BCVA of 0.2 decimals and a flat retina.

Patient 2 was a 51-year-old female who presented with symptoms suggestive of right-eye RRD. On examination, superior temporal RRD was confirmed. Urgent PPV was arranged the following day, and the operative findings included superior-temporal RRD from 11–7 o'clock, with a massive horseshoe tear located at the 11 o'clock position. The macula was removed. An intraoperative endolaser was applied 360˚ around the tear. SO was injected with good filling.

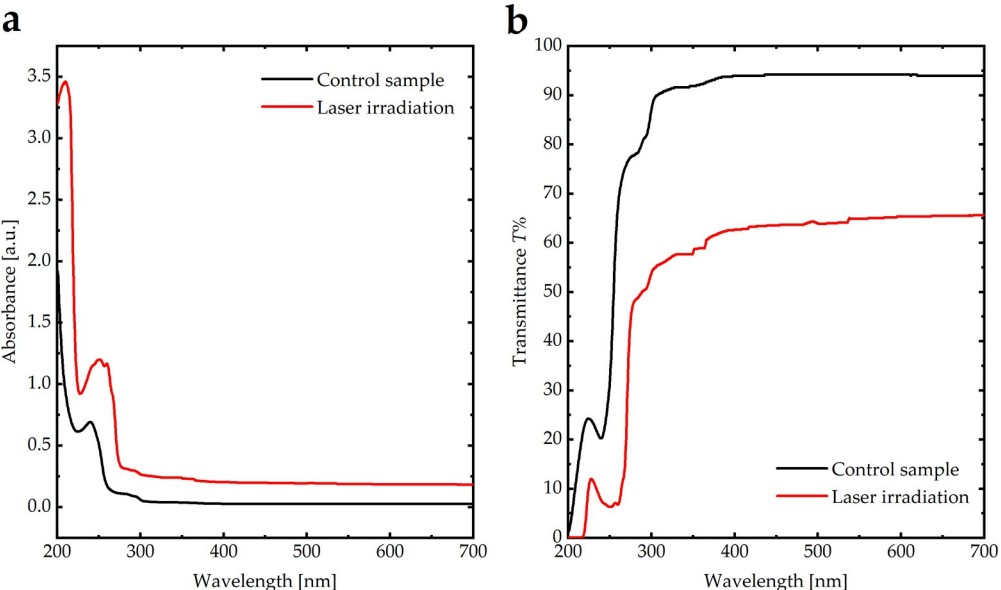

**Fig 5.** **(a)** Absorbance and **(b)** transmittance spectra of pure SO and SO under laser irradiation.

One month after PPV, the patient achieved a BCVA of 0.2 decimals with a flat retina under SO. However, another tear was found superior temporally, and she underwent three sessions of argon laser retinopexy. Twelve months later, the patient exhibited decreased vision. The BCVA was 0.05 with mild cataracts and flat retinas under SO. The SO appeared turbid and emulsified. Accordingly, SO removal and cataract extraction were performed without complications. During the operation, the SO was found to be severely emulsified. One month later, the patient presented with a BCVA of 0.2 decimals and a flat retina.

*Patient 3 versus 4.* Patient 3 was a 41-year-old male; he presented with symptoms suggestive of right-eye RRD. On examination, superior temporal RRD was confirmed. Urgent PPV was arranged on the same day, and the operative findings included superior temporal RRD from 10–3 to o'clock with multiple tears located at the 10, 6, 5, 4, and 1 o'clock positions, along with severe peripheral lattice and cystic changes. The macula was attached during surgery. An intraoperative endolaser was applied around the tear and extended 360˚. SO was injected with good filling. After 12 months, the SO was removed from the right eye via cataract extraction without complications. One month later, the patient presented with a BCVA of 0.9 decimals and a flat retina.

Patient 4 was a 46-year-old male who presented with symptoms suggestive of RRD in the left eye. On examination, superior temporal RRD was confirmed. Urgent PPV was arranged on the same day, and the operative findings included superior temporal RRD from 1–4 o'clock

**Table 2. General demographics and characteristics.**

| Patient # | Sex | Age [Years] | Comorbidities | Laterality | Primary Pathology |
|---|---|---|---|---|---|
| Patient 1 | Female | 47 | Medically free | Left | RRD |
| Patient 2 | Female | 51 | Hypertension | Right | RRD |
| Patient 3 | Male | 41 | Medically free | Right | RRD |
| Patient 4 | Male | 46 | Medically free | Left | RRD |

Abbreviations: RRD: rhegmatogenous retinal detachment

**Table 3. Operative and technical details.**

| Patient # | Primary operation | Duration of SO within the eyes [months] | SOR Operation | Argon laser exposure before SOR | Anesthesia | Posterior Segment Status post SOR |
|---|---|---|---|---|---|---|
| Patient 1 | PPV | 19 | SOR with cataract extraction | No | General | Flat |
| Patient 2 | PPV | 12 | SOR with cataract extraction | Yes | Local | Flat |
| Patient 3 | PPV | 12 | SOR with cataract extraction | No | General | Flat |
| Patient 4 | PPV | 9 | SOR with cataract extraction | Yes | Local | Flat |

Abbreviations: PPV: pars plana vitrectomy; SOR: silicone oil removal

with multiple tears located at 1, 11, and 9 o'clock positions. The macula was then attached. An intraoperative endolaser was applied and extended 360˚ around the tear. SO was injected with good filling. One month after PPV, the patient achieved a BCVA of 0.7 decimals with a flat retina under silicone oil. However, other tears were found at the periphery of all quadrants, and he underwent an argon laser retinopexy session. After nine months, the patient presented with a reduction in visual acuity. The BCVA was 0.3 decimals with mild cataracts and flat retina under emulsified SO. SO removal and cataract extraction were performed in the left eye without complications. During operation, the SO was emulsified. One month later, the patient presented with a BCVA of 0.4 decimals and a flat retina.

**3.4.2 Physical and chemical properties of patients' SO samples.** The change in the chemical structure of SOs between a new, unused SO (control) sample and the samples extracted from the four patients was investigated by analyzing the vibrational bands in the FTIR spectra (Fig 6A). The SO control sample exhibited different vibrational bands based on its chemical structure: i.e., Si–O (680 cm$^{-1}$), Si–(CH$_3$)$_2$ (790 cm$^{-1}$), Si–(CH$_3$)$_3$ (870 cm$^{-1}$), Si–

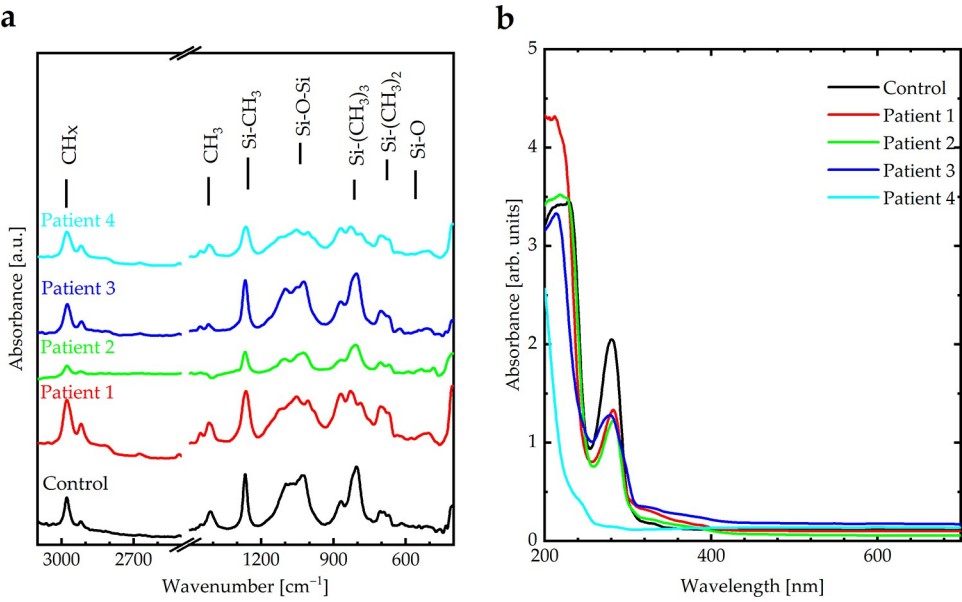

**Fig 6. (a)** FTIR and **(b)** absorbance spectra of SO for the four patients compared to the control silicon sample (pure silicone oil).

O–Si (1,010 cm$^{-1}$), Si–CH$_3$ (1,260 cm$^{-1}$), CH$_3$ (1,405 cm$^{-1}$), and CH$_x$ (2,960 cm$^{-1}$). Si–O and Si–O–Si bonds are the main components of the skeleton. If the primary bonds are broken, oxygen will have a chance to connect with other elements, such as C or H, and Si will have the opportunity to bond with H. A closer examination of the Si–O and Si–O–Si bonds revealed a decrease in intensity for all patients, indicating the breakdown of bonds and emulsification. However, the Si–O and Si–O–Si vibrational bands of patients 2 and 4 exhibited a more significant reduction, which may be attributed to the exposure of these samples to argon laser retinopexy, as discussed in Section 3.3.

Furthermore, the viscosity of the control SO was 5,394 mPa.s, which decreased to 5,152; 4,894; 5,207; and 4,914 mPa.s for Patients 1–4, respectively. This may be attributed to the presence of emulsification. However, Patients 2 and 4 had lower viscosities than Patients 1 and 3, which was attributed to exposure to argon laser retinopexy.

Fig 6B shows the absorbance spectra of the newly unused SO (control sample) and the samples extracted from the four patients. The absorbance values of the control samples in the visible region were lower than those of the patient samples. This observation is consistent with the emulsification process, which produces free radicals and broken bonds, resulting in dangling bonds free to interact with incident light and absorb its energy. The absorbance band at 282 nm for the control sample represents the absorption band of the SO backbone. This band decreased in patient samples, which can be attributed to emulsification. However, Patients 2 and 4 had a higher variation in the absorbance bands than patients 1 and 3, which was attributed to exposure to argon laser retinopexy.

## 4. Discussion and conclusions

To the best of our knowledge, this is the first *in vivo* and *in vitro* study to assess the optical, chemical, and physical behavior of unused SO samples exposed to different conditions and to correlate these findings with the behavior of SO samples obtained from the vitrectomized eyes of patients who underwent PPV for RRD. This study showed that UV light, heat, microwave radiation, and aging do not influence the properties of SO. Furthermore, the study showed that exposure to argon laser retinopexy may increase the rate of SO emulsification by facilitating bond cleavage, as indicated by FTIR analysis. This resulted in decreased transmittance, increased absorbance, and decreased viscosity. These effects were observed in SO samples exposed to a similar argon laser retinopexy.

SO emulsification has many complications, including aggravation of PVR, increased failure of anatomical success, refractory glaucoma, band keratopathy, and an increased risk of inflammation [7]. Several factors influence the tendency of SO to emulsify, including interfacial surface tension, content of low molecular weight compounds, viscosity, chemical structure, and absorption of several biological substances from intraocular fluids and tissues (known as emulsifiers) [20]. The effects of these factors, such as blood, inflammatory biomarkers, and lipids, are driven by diminishing surface tension and emulsification [20, 21]. Moreover, it has been postulated that certain mechanical factors, such as shear force during aqueous and oil movements and rapid eye movements, can cause emulsification [21, 22]. Moreover, the viscosity and molecular weight may affect the emulsification rate. Less viscous silicone oils (lower molecular weight) emulsify earlier than more viscous oils [10, 22]. Furthermore, the duration of SO implantation correlates with the development of emulsification; the longer the duration, the higher the rate of emulsification [23]. Our study revealed that aging did not influence SO emulsification. Other factors that may contribute to emulsification include the use of perfluorocarbon liquid in direct exchange with SO, the presence of impurities in the SO, and higher mechanical forces from intraocular instruments [24].

The mean onset duration of SO emulsification was reported to be 13.2 ± 4.8 months (ranging from 5–24 months). Many hypotheses have been proposed to explain the mechanism of SO emulsification. The shearing forces induced by saccadic eye movements separate tiny SO droplets from the central SO bubbles [25]. The relative motion between the SO molecules and the vitreous cavity wall arises because the liquids in the vitreous cavity do not entirely follow eye movements. The viscosity of any liquid determines the amount of relative motion between the liquid and the eye. High-viscosity liquids have a more extraordinary ability in momentum diffusion, with a subsequent higher extent of the liquid following eye movement [25]. In addition, the greater the relative motion between the SO molecules and the vitreous cavity, the higher the relative velocity, with subsequently higher shear stress acting on the SO/aqueous interface, which triggers emulsification. As the more viscous SO tends to have less relative motion against eye movement, the resulting shear stress acting on the SO/aqueous interface is lower than that of the less viscous SO, with a subsequent reduction in emulsification.

A new and advanced method for avoiding these complications is using a foldable capsular vitreous body. It is a novel vitreous substitute composed of a thin (30 μm) vitreous-shaped capsule, a drain tube, and a valve [26–28]. Compared with SO, the foldable capsular vitreous body has a smaller optical effect on refraction [26]. Furthermore, foldable capsular vitreous bodies can provide sustained release of dexamethasone, serve as an intravitreal drug delivery system, and have a relatively lower overall postsurgical complication rate than the SO tamponade, with no need for patients to maintain a prone position after surgery [26].

One factor that may induce emulsification is the effect of ophthalmic laser exposure in vitrectomized patients with SO tamponade [7]. A laser can emit a coherent, robust, monochromatic beam of electromagnetic radiation. Coherent radiation indicates that all photons of specific wavelengths are in phase with each other with limited divergence. Monochromatic radiation is a single-frequency wavelength with minimal chromatic aberration. Lasers have many effects on tissues, including photochemical, photocoagulation, and photomechanical effects. In the photocoagulation effect, the temperature of exposed tissue is elevated from 37°C to at least 100°C, resulting in the denaturation of tissue structures and protein and coagulation at the exposed tissue site. This results from the conversion of light into heat. The xanthophyll, melanin in the macula, and hemoglobin absorb light from the laser. Theoretically, green argon, krypton red, and diode lasers can be used in retinal surgery. The green argon laser is the principal laser used in retinal surgery. It is absorbed mainly by hemoglobin pigments, retinal pigment epithelium (RPE), choriocapillaris, and a layer of rods and cones. The melanin granules readily absorb it, and it coagulates from choriocapillaris to the inner nuclear layer of the retina [29–32].

SO contains a long chain of linked monomers, each containing a silicon atom sharing two oxygen atoms at the main wings and $CH_3$ molecules at the other two accessible peripherals. The linear long chain consists of repeated $O = Si = O$ segments that must be strong enough to maintain a robust polymeric chain with a high molecular weight, such as in the SO used in ophthalmic practice.

According to the FTIR spectra, SO exhibited different vibrational bands based on its chemical structure: Si–O (680 cm$^{-1}$), Si–$(CH_3)_2$ (790 cm$^{-1}$), Si–$(CH_3)_3$ (870 cm$^{-1}$), Si–O–Si (1,010 cm$^{-1}$), Si–$CH_3$ (1,260 cm$^{-1}$), $CH_3$ (1,405 cm$^{-1}$), and $CH_x$ (2,960 cm$^{-1}$). The FTIR spectra of SO under thermal treatment, UV irradiation (long and short), microwave irradiation, visible light irradiation, and after aging were identical to those of untreated SO, indicating the absence of broken bonds; therefore, no significant emulsification occurred. Additionally, the optical absorbance and viscosity of SO under these conditions remained similar to those of the control sample, indicating that the optical properties and viscosity of SO were stable. Conversely, SO exposed to the laser demonstrated an apparent effect on Si–O and Si–O–Si bonds indicative of

broken bonds; therefore, significant emulsification occurred. This suggests that bond cleavage and emulsification in everyday life may be attributed to vital operations within the eye, such as exposure to laser irradiation. In addition, the optical absorbance and viscosity exhibited significant variations compared to those of the SO untreated by laser irradiation. Finally, four patients, two of whom were treated with laser exposure and the other two who were not treated with laser irradiation, were compared to validate previous results. The results suggest that patients exposed to laser irradiation exhibited more emulsification of SO within the eyes than those who were not treated, with a lower duration of SO. Adequate intraoperative endolaser use during PPV should be promptly performed. More extensive studies should be conducted to validate these results. Future studies should investigate these biological factors. Moreover, a larger trial comparing the SO to the foldable capsular vitreous body may provide valuable insights.

This study has some limitations. First, the small sample size was a major limitation. Second, no statistical tests were used in this study, which may have limited the validity of its findings. Third, this study focused on the physical effect of SO, but it did not consider the role of inflammatory biomarkers and other biological materials that may develop in the operated eye and potentially influence the emulsification of SO.

## Supporting information

**S1 File. FTIR analysis data.**
(XLSX)

**S2 File. Optical analyses data.**
(CSV)

## Author Contributions

**Conceptualization:** Rami Al-Dwairi, Ahmad. A. Ahmad, Areen Bani-Salameh, Seren Al Beiruti, Abdulrawof Alhabachi.

**Data curation:** Rami Al-Dwairi, Ahmad. A. Ahmad, Abdelwahab Aleshawi, Seren Al Beiruti, Abdulrawof Alhabachi.

**Formal analysis:** Abdelwahab Aleshawi, Qais M. Al-Bataineh.

**Funding acquisition:** Rami Al-Dwairi.

**Investigation:** Rami Al-Dwairi, Ahmad. A. Ahmad, Qais M. Al-Bataineh, Areen Bani-Salameh.

**Methodology:** Rami Al-Dwairi, Ahmad. A. Ahmad, Abdelwahab Aleshawi, Areen Bani-Salameh, Ihsan A. Aljarrah, Seren Al Beiruti, Abdulrawof Alhabachi.

**Project administration:** Rami Al-Dwairi, Ahmad. A. Ahmad.

**Resources:** Qais M. Al-Bataineh, Ihsan A. Aljarrah.

**Software:** Qais M. Al-Bataineh, Areen Bani-Salameh, Ihsan A. Aljarrah.

**Supervision:** Rami Al-Dwairi, Ahmad. A. Ahmad, Qais M. Al-Bataineh.

**Validation:** Rami Al-Dwairi, Ahmad. A. Ahmad, Abdelwahab Aleshawi, Ihsan A. Aljarrah, Seren Al Beiruti, Abdulrawof Alhabachi.

**Visualization:** Rami Al-Dwairi.

**Writing – original draft:** Rami Al-Dwairi, Ahmad. A. Ahmad, Abdelwahab Aleshawi, Qais M. Al-Bataineh.

**Writing – review & editing:** Areen Bani-Salameh, Ihsan A. Aljarrah, Seren Al Beiruti, Abdulrawof Alhabachi.

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
