## [Decision Letter · Decision Letter 0]

4 Nov 2024

PONE-D-24-26615

The Influence of UV-Visible Light, Microwave Radiation, Argon Laser, Heating and Aging Process on Silicone Oil Utilized as Intravitreal Implants: Experimental Exposure with Clinical Correlation

PLOS ONE

Dear Dr. Al-Dwairi,

Thank you for submitting your manuscript to PLOS ONE. After careful consideration, we feel that it has merit but does not fully meet PLOS ONE’s publication criteria as it currently stands. Therefore, we invite you to submit a revised version of the manuscript that addresses the points raised during the review process.

We look forward to receiving your revised manuscript.

Kind regards,

Ayman Elnahry

Academic Editor

PLOS ONE

“The authors would like to acknowledge Jordan University of Science and Technology for supporting the research projects validated by the Deanship of Scientific Research (Funding ID: 20230529)”

Additional Editor Comments:

The manuscript idea is novel but the manuscript is poorly written. It needs to be significantly shortened and repetitive statements removed. The English language needs extensive editing, perhaps using a professional service. Limitations of the study such as sample size and lack of statistical analysis should be mentioned and recommendations for future studies stated. Please respond fully to reviewers’ criticism/comments. Discussion section which is mentioned as “Conclusions” is very short and relevant literature needs to be discussed. The value of having SO in foldable capsular vitreous body can be briefly discussed.

Reviewers' comments:

Reviewer's Responses to Questions

**Comments to the Author**

1. Is the manuscript technically sound, and do the data support the conclusions?

Reviewer #1: Partly

Reviewer #2: Yes

Reviewer #3: Partly

2. Has the statistical analysis been performed appropriately and rigorously? 

Reviewer #1: N/A

Reviewer #2: N/A

Reviewer #3: No

3. Have the authors made all data underlying the findings in their manuscript fully available?

Reviewer #1: Yes

Reviewer #2: Yes

Reviewer #3: Yes

4. Is the manuscript presented in an intelligible fashion and written in standard English?

Reviewer #1: No

Reviewer #2: Yes

Reviewer #3: No

5. Review Comments to the Author

Reviewer #1: Running title: Silicone oil under different condition

- I would suggest to add an “s” to condition.

- A thorough English Editing is compulsory.

Page 4. Line 75

Many hypotheses could explain the incompletely understood mechanism of emulsification of SO.

- Rephrase this sentence because it is not clear

Page 6 line 115: Aging factors.

The Authors should explain in detail how this aging process has been done and what are the aging factors.

Page 10 line 224-226; The Authors attested that: To simulate the SO inside the eye while considering the aging factor, we mixed SO with BSS and aqueous humor.

- Unfortunately, I do not agree with this postulation as in the vitreous chamber after surgery a lot of inflammatory cytokines and pro inflammatory factors can be released. Think only to the blood elements that can be released into the vitreous chamber.

Page 17 lines 368-372; The Authors attested: The FTIR spectra of SO under thermal treatment, U.V. irradiation (long and short), microwave irradiation, visible irradiation, and aging exhibit identical spectra for untreated SO, indicating that there are no broken bonds and; therefore, no significant emulsification occurred. Additionally, the optical absorbance and the viscosity of SO under these parameters remain almost the same as the control sample, with slight changes, indicating that the optical properties and viscosity are stable under these parameters.

- I do agree with this statement from physical point of view, but unfortunately in the vitreous chamber there are also al lot of biological components that must be considered before make any conclusion.

ABSTRACT.

The Authors also concluded that : “ It was concluded that the patient's samples treated with argon laser exposure exhibited more emulsification than the other two, which had a lower duration of SO within the eyes.”

- So what the Authors suggest? To avoid laser when SO is in the vitreous chamber or an early removal after laser?

Page 18 Line 379-381

The Author attested: Finally, four patients, two of whom were treated with laser exposure and the other two patients who were untreated with laser irradiation, were compared to validate the previous results. It can be concluded that the patients' samples that were treated with laser exposure exhibited more emulsification than the other two patients who were not treated with a lower duration of SO within the eyes.

- This effect could also be related to the interaction with inflammatory cytokines released after laser treatment. These considerations must be considered from the Authors. Not only physics but biological interactions of SO.

Did the Authors measure the proteins concentration in removed SOs?

Reviewer #2: Novel study combining in vivo and in vitro studies using advanced technology. The authors need to acknowledge that it is difficult to confirm based on single case the effect of silicone oil emulsification. For example, the eye that had argon Laser showed the most difference as concluded by the authors, but this eye could have had PVR or inflammation, more activity of the patient, more sun exposure or less complete oil fill, etc..These variables (inflammation, more activity of the patient, more sun exposure or less complete oil fill) need to be mentioned as drawbacks of the study besides the small number of patients. Otherwise this is a good start for larger studies.

Reviewer #3: I admire the novelty of the idea and the translational value of this work, however, there are some shortcomings that affect the integrity of the conclusions made by the manuscript in its current status:

1. Statistical analysis is needed to prove the physical changes to silicone oil is related to the tested factors and avoid type I error. This is not possible with the current low number of samples. I believe more oil samples are needed to consolidate those observations.

2. More details needed in the study design and more background knowledge. For example, background knowledge on FTIR would be of benefit.

3. I would advise on a more relevant literature review to publications with similar work.

4. The manuscript might as well benefit from language revisions.

6. PLOS authors have the option to publish the peer review history of their article (what does this mean?). If published, this will include your full peer review and any attached files.

Reviewer #1: No

Reviewer #2: No

Reviewer #3: No

---

## [Author Response · Author response to Decision Letter 0]

7 Nov 2024

Academic Editor

PLOS ONE

Dear editor,

My self and the co-authors are pleased to resubmit the manuscript entitled ‘The Influence of UV-Visible Light, Microwave Radiation, Argon Laser, Heating and Aging Process on Silicone Oil Utilized as Intravitreal Implants: Experimental Exposure with Clinical Correlation, ID: PONE-D-24-26615” to be considered for publication in PLOS ONE. The revised manuscript takes into consideration both the editorial and reviewers’ comments. Kindly find below a point-by-point response to those comments, along with an uploaded copy marked with MS track changes indicating changes to the manuscript.

We would like to thank your editorial team and the reviewers for their time and efforts in reviewing our manuscript. 

Editorial comments 

Thank you so much for your efforts and time in improving this manuscript. Your comments are much appreciated, and the manuscript will be modified accordingly,

1. The manuscript idea is novel but the manuscript is poorly written. It needs to be significantly shortened and repetitive statements removed. The English language needs extensive editing, perhaps using a professional service

Response: Professional English language and scientific editing was performed. Thank you so much.

2. Limitations of the study such as sample size and lack of statistical analysis should be mentioned and recommendations for future studies stated.

Response: Thank you. The small sample size with lack of statistical analysis was added to the limitations. Recommendations were amended to the revised manuscript. Thank you so much.

3. Discussion section which is mentioned as “Conclusions” is very short and relevant literature needs to be discussed. The value of having SO in foldable capsular vitreous body can be briefly discussed. 

Response: Thank you so much. The discussion was combined with the results. More relevant literature review was discussed in both sections. In addition, the value of SO in foldable capsular vitreous body was discussed. Thank you so much for efforts in improving the manuscript.

Journal Requirements

Response: Thanks so much. It was modified according to the template. 

2. Please state what role the funders took in the study.

Response: Thank you so much. The funders had no role in study design, data collection and analysis, decision to publish, or preparation of the manuscript. It was added to the revised manuscript.

Response: Thank you so much. We have modified the statement into “Data are available upon request from the corresponding author. The data collection sheets used are available in the appendix for use”. In addition, we have uploaded supplementary material for the data.

4. PLOS requires an ORCID iD for the corresponding author in Editorial Manager. 

Response: Thank you. We have added the ORCID.

Response: Thank you. We amended the modifications. 

Comments from Reviewer #1:

Many thanks for your time and efforts in improving our manuscript. Much appreciated

1. The Running title: Silicone oil under different condition. I would suggest to add an “s” to condition.

Response: Thank you so much. It was modified. 

2. A thorough English Editing is compulsory.

Response: Thank you. We have modified the manuscript and extensive language editing was performed.

3. Many hypotheses could explain the incompletely understood mechanism of emulsification of SO. - Rephrase this sentence because it is not clear.

Response: Thank you for your respectful point. It was rephrased and modified in the revised manuscript. 

4. age 6 line 115: Aging factors. The Authors should explain in detail how this aging process has been done and what are the aging factors.

Response: Thank you very much for this important point. The aging process means the effect of duration on the emulsification of SO. It was hypothesized that the longer duration of SO in the vitreous cavity may increase the rate of emulsification. Accordingly, to simulate the SO inside the eye while considering the aging factor, we mixed the SO with a balanced salt solution (BSS) and eye aqueous humor. We studied their properties after aging them for 6 and 12 months, respectively. However, this a partial model, as you stated, many inflammatory markers and cytokines may influence the SO emulsification, but technically were difficult to be added to the SO. The required modifications were added to the revised manuscript with more clarifications.

5. Page 10 line 224-226; The Authors attested that: To simulate the SO inside the eye while considering the aging factor, we mixed SO with BSS and aqueous humor.

- Unfortunately, I do not agree with this postulation as in the vitreous chamber after surgery a lot of inflammatory cytokines and pro inflammatory factors can be released. Think only to the blood elements that can be released into the vitreous chamber.

Response: Thank you very much for this valid point. As aforementioned comment, we have tried to stimulate the aging or time factor as much as possible, however, it would be difficult to extract the exact vitreous inflammatory markers and add them to the SO. We mix the SO with BSS and aqueous. The aqueous was extracted at the time of SO removal in order to resemble the vitreous with its inflammatory components as much as possible. Accordingly, we have tried a partial model to stimulate the process. We have clarified this point and added to the limitations. Thank you so much.

6. Page 17 lines 368-372; The Authors attested: The FTIR spectra of SO under thermal treatment, U.V. irradiation (long and short), microwave irradiation, visible irradiation, and aging exhibit identical spectra for untreated SO, indicating that there are no broken bonds and; therefore, no significant emulsification occurred. Additionally, the optical absorbance and the viscosity of SO under these parameters remain almost the same as the control sample, with slight changes, indicating that. the optical properties and viscosity are stable under these parameters.

- I do agree with this statement from physical point of view, but unfortunately in the vitreous chamber there are also al lot of biological components that must be considered before make any conclusion.

Response: Thank you for this point. Yes, we agree with your point, accordingly, we have modified the Conclusions and amended your point regarding the biological effects.

7. The Authors also concluded that : “ It was concluded that the patient's samples treated with argon laser exposure exhibited more emulsification than the other two, which had a lower duration of SO within the eyes.”

- So what the Authors suggest? To avoid laser when SO is in the vitreous chamber or an early removal after laser?.

Response: Thank so much this valid comment. In certain circumstances the additional postoperative laser in unavoidable. However, out suggested educational purpose is to perform adequate and comprehensive endo-laser treatment during the pars plana vitrectomy and before SO implantation. We have amended this education point.

8. The Author attested: Finally, four patients, two of whom were treated with laser exposure and the other two patients who were untreated with laser irradiation, were compared to validate the previous results. It can be concluded that the patients' samples that were treated with laser exposure exhibited more emulsification than the other two patients who were not treated with a lower duration of SO within the eyes.

- This effect could also be related to the interaction with inflammatory cytokines released after laser treatment. These considerations must be considered from the Authors. Not only physics but biological interactions of SO.

Did the Authors measure the proteins concentration in removed SOs?. 

Response: Thank you very very much. The conclusion was made from the experimental in-vitro exposure of naïve unused SO sample to the green argon laser. However, you suggestion is very important and cannot be avoided. The biological factors may have critical effects on SO emulsification. We have tried to approach the biological effect by mixing the SO with aqueous extracted at the time of SO removal. The protein was not measured as the size of the SO was too small to perform enough physical and biological tests. We have amended this point to the limitations and modified the revised manuscript. 

Comments from Reviewer #2: 

Thank you for efforts in reviewing our manuscript. 

1. Novel study combining in vivo and in vitro studies using advanced technology. The authors need to acknowledge that it is difficult to confirm based on single case the effect of silicone oil emulsification. For example, the eye that had argon Laser showed the most difference as concluded by the authors, but this eye could have had PVR or inflammation, more activity of the patient, more sun exposure or less complete oil fill, etc..These variables (inflammation, more activity of the patient, more sun exposure or less complete oil fill) need to be mentioned as drawbacks of the study besides the small number of patients. Otherwise this is a good start for larger studies.

Response: Thank you so much for your encouraging and respectful comment. We agree with your that important biological and environmental factors should be taken into consideration. Actually, some eyes that needed postoperative laser were more active and complicated eyes. However, the findings of in-vitro change of SO characteristics after exposure of unused SO sample to argon laser supported this suggestion. These drawbacks were added to the limitations. Thank you again and again. 

Comments from Reviewer #3: 

Thank you for efforts in reviewing our manuscript. Your comments and points are much appreciated and encouraging to us.

1. Statistical analysis is needed to prove the physical changes to silicone oil is related to the tested factors and avoid type I error. This is not possible with the current low number of samples. I believe more oil samples are needed to consolidate those observations.

Response: Thank you very much. Yes, we agree with your respectful comment. The cause for the small sample size the preciosity of extracted SO samples along with high-cost tests. However, we are working on a much larger sample size. This point was amended in the limitations. Thank you so much.

2. More details needed in the study design and more background knowledge. For example, background knowledge on FTIR would be of benefit.

Response: Thank you so much for your meaningful point. We have amended more background knowledge in the Methods. 

3. I would advise on a more relevant literature review to publications with similar work.

Response: Thank you so much. We have elaborated more in the Discussion with more literature reviews. 

4. The manuscript might as well benefit from language revisions.

Response: Thank you very much. Extensive language revisions were made. Thank you again.

Again, we thank the editor and reviewers for their constructive comments. We hope that we have sufficiently addressed their concerns.

Sincerely,

Authors

---

## [Editor Report · Decision Letter 1]

18 Nov 2024

PONE-D-24-26615R1The Influence of UV-visible light, microwave radiation, argon laser, heating and aging process on silicone oil utilized as intravitreal implants: Experimental exposure with clinical correlationPLOS ONE

Dear Dr. Al-Dwairi,

Thank you for submitting your manuscript to PLOS ONE. After careful consideration, we feel that it has merit but does not fully meet PLOS ONE’s publication criteria as it currently stands. Therefore, we invite you to submit a revised version of the manuscript that addresses the points raised during the review process.

We look forward to receiving your revised manuscript.

Kind regards,

Ayman Elnahry

Academic Editor

PLOS ONE

Journal Requirements:

Additional Editor Comments:

Thanks for submitting your revised manuscript.

Unfortunately professional English language editing was not performed. The English language still needs extensive editing and correction of spelling mistakes. For example, one of your subtitles reads “partients and design”. I suggest to use a professional English language editor to improve the style of the manuscript.

---

## [Author Response · Author response to Decision Letter 1]

22 Nov 2024

Academic Editor

PLOS ONE

Dear editor,

My self and the co-authors are pleased to resubmit the revised version of the manuscript entitled ‘The influence of UV-visible light, microwave radiation, argon laser, and heating and aging processes on silicone oil utilized as intravitreal implants: Experimental exposure with clinical correlation, ID: PONE-D-24-26615R1” to be considered for publication in PLOS ONE. The revised manuscript takes into consideration all editorial comments. Kindly find below a point-by-point response to those comments, along with an uploaded copy marked with MS track changes indicating changes to the manuscript.

Again and again, many thanks for our respectful editor and the editorial staff for their efforts. Your efforts are much appreciated and valuable and will be kept in heart

Editorial comments 

Thank you so much for your efforts and time in improving this manuscript. 

“Thanks for submitting your revised manuscript. Unfortunately professional English language editing was not performed. The English language still needs extensive editing and correction of spelling mistakes. For example, one of your subtitles reads “partients and design”. I suggest to use a professional English language editor to improve the style of the manuscript.”

Response: Thank you so much for your efforts and the comments. We have sent the manuscript for a professional English language editing service and we have uploaded the certificate. Thank you again

Journal Requirements

Response: Thanks so much. We have revised and modified the reference list. Furthermore, to cooperate with your efforts, we have included two references lists, one without software and one arranged with the software. 

Again, we thank the editor and reviewers for their constructive comments. We hope that we have sufficiently addressed their concerns.

Sincerely,

Authors

---

## [Editor Report · Decision Letter 2]

9 Dec 2024

The influence of UV-visible light, microwave radiation, argon laser, and heating and aging processes on silicone oil utilized as intravitreal implants: Experimental exposure with clinical correlation

PONE-D-24-26615R2

Dear Dr. Al-Dwairi,

We’re pleased to inform you that your manuscript has been judged scientifically suitable for publication and will be formally accepted for publication once it meets all outstanding technical requirements.

Kind regards,

Ayman Elnahry

Academic Editor

PLOS ONE

Additional Editor Comments (optional):

I thank the authors for completing the requested revisions.
---

## [Editor Report · Acceptance letter]

11 Dec 2024

PONE-D-24-26615R2 

PLOS ONE

Dear Dr. Al-Dwairi, 

I'm pleased to inform you that your manuscript has been deemed suitable for publication in PLOS ONE. Congratulations! Your manuscript is now being handed over to our production team.

Kind regards, 

on behalf of

Dr. Ayman Elnahry 

Academic Editor

PLOS ONE